# Fortifying the Foundations: A Comprehensive Approach to Enhancing Mental Health Support in Educational Policies Amidst Crises

**DOI:** 10.3390/healthcare11101423

**Published:** 2023-05-14

**Authors:** Christian J. Wiedermann, Verena Barbieri, Barbara Plagg, Pasqualina Marino, Giuliano Piccoliori, Adolf Engl

**Affiliations:** 1Institute of General Practice and Public Health, Claudiana—College of Health Professions, 39100 Bolzano, Italybarbara.plagg@am-mg.claudiana.bz.it (B.P.);; 2Department of Public Health, Medical Decision Making and Health Technology Assessment, University of Health Sciences, Medical Informatics and Technology, 6060 Hall, Tyrol, Austria; 3Faculty of Education, Free University of Bolzano, 39100 Bolzano, Italy

**Keywords:** mental health, children and adolescents, school-based interventions, coping skills

## Abstract

In recent times, global crises such as the COVID-19 pandemic, climate change, and geopolitical conflicts have significantly impacted pupils’ mental health. This opinion article presents evidence-based recommendations to bolster mental health support within educational systems, aiming to alleviate the psychological burden faced by students during these challenging times. This article argues that a proactive, holistic approach to mental health is essential for building a resilient educational infrastructure. More than ever, we support the call for the integration of mental health education into the core curriculum, equipping students with vital coping skills and fostering emotional intelligence. Additionally, we emphasize the importance of training educators and staff to identify and address mental health issues. Furthermore, this article highlights the need for interdisciplinary collaboration involving general practitioners, mental health professionals, community organizations, and policymakers in crafting and implementing support strategies. Educational institutions can effectively leverage the expertise of diverse stakeholders to create targeted interventions by cultivating partnerships. Finally, the significance of continuously evaluating and refining mental health support policies to ensure their efficacy and adaptability in the face of evolving crises is emphasized. Through these comprehensive recommendations, this opinion article seeks to catalyze a transformation in educational policies, prioritize mental health support, and empower pupils to thrive during tumultuous times.

## 1. Introduction

Global crises, such as the Coronavirus Disease 2019 (COVID-19) pandemic, climate change, and geopolitical conflicts, have significantly impacted the mental health of children and adolescents. The COVID-19 pandemic has disrupted education and led to increased stress, anxiety, and other mental health issues among pupils [1,2,3,4]. Climate-change-related anxiety, or “eco-anxiety,” is becoming more prevalent as people worry about the future of the planet [5,6,7]. Youths in conflict-affected areas face unique mental health challenges due to trauma, displacement, and instability [8,9,10,11].

The mental health of youth has been a concern since the late 1990s, and issues such as COVID-19 and ongoing climate change, war, energy crisis, and socioeconomic stress are now thought to exacerbate the situation. To target preventive and therapeutic interventions, identifying and defining each factor, such as climate change, war, and pandemics, is necessary to understand how they can impact mental health individually and collectively. To develop focused interventions for prevention, it is important to differentiate these factors. Conducting research and analyzing existing data to determine the specific impact of each risk factor on mental health is a prerequisite.

Categorizing affected children and adolescents based on their specific risks, geographical locations, socioeconomic backgrounds, and other relevant factors can help tailor interventions to address the unique needs of each group [12,13,14]. Preventive interventions may include youth education on climate issues and empowerment to take action [15]. Support groups can be created to discuss eco-anxiety and coping mechanisms [16,17]. Psychological support and trauma counseling can be provided for those directly affected or displaced by war [18,19], and peace education and conflict resolution programs can be provided [20,21]. Offering financial literacy and job training programs to help youth cope with economic stress and uncertainty represents another preventive possibility [22,23].

Schools play a crucial role in providing safety and mental health assessments and interventions for children, especially in the context of disasters and crises, making them a critical setting for the delivery of mental health support services [24]. Furthermore, tailored, school-based interventions have proven effective in providing educational and psychosocial support for conflict-affected youths, addressing issues such as academic underachievement and its impact on mental health [25]. This opinion article presents evidence-based recommendations to bolster mental health support within educational systems, aiming to alleviate the psychological burden faced by youths during these challenging times.

## 2. Methods

Our research problem centered on the pressing need for improved mental health support within educational systems, exacerbated by global crises, such as the COVID-19 pandemic, climate change, and geopolitical conflicts. We aimed to present evidence-based recommendations to bolster mental health support within these systems.

To address this problem, our research procedure involved an extensive review and synthesis of relevant literature, covering topics such as the impact of global crises on students’ mental health, effective mental health support strategies within education, and the role of interdisciplinary collaboration in mental health support. We critically analyzed these sources to develop our recommendations, ensuring that they were grounded in evidence and aligned with international standards. Additionally, our procedure involved seeking expert opinions from educators, mental health professionals, and policymakers to gain diverse perspectives on the issue and to ensure the practical applicability of our recommendations.

## 3. Prioritization of Mental Health in Schools

A child’s development is influenced by various environments, including the family, school, and community. While families are vital in supporting children’s mental health during periods of crisis, schools play a complementary and indispensable role in promoting mental well-being. To ensure comprehensive support, it is essential to address mental health issues across all contexts. Schools, as primary environments where children spend a significant amount of time, have a unique opportunity to contribute to this support system [26].

A holistic approach to mental health in schools considers the whole person, addressing not only cognitive development, but also emotional, social, and psychological well-being [27]. This approach fosters the growth of well-rounded individuals who are better prepared for life beyond school. It is essential to build a resilient educational infrastructure that supports academic success through early intervention, whole-person development, inclusivity, resilience, de-stigmatization, staff well-being, and stronger communities [28,29,30]. Inclusive learning environments that acknowledge and accommodate diverse needs contribute beneficially to creating an environment where all pupils, including those with mental health challenges, feel welcomed, supported, and empowered to succeed [31,32,33]. Such an approach equips children and adolescents with skills to cope with stress, adapt to change, and bounce back from adversity, which are essential life skills that contribute to long-term success.

Good mental health is a foundation for educational and academic success [34,35,36]. Proactively addressing mental health issues helps identify and address potential problems before escalating. Early intervention can prevent more severe mental health challenges, leading to improved outcomes for pupils and reducing the burden on educational systems [37]. By promoting mental health and well-being, a proactive and holistic approach helps students develop resilience. 

Proactively addressing mental health issues in educational settings can help to reduce the stigma associated with mental health challenges [12,13,38]. This fosters a more open and supportive culture that encourages students to seek help when needed. The focus on mental health also extends to educators and support staff [39]. By fostering mental well-being [40], educational infrastructure has become more resilient and better equipped to support pupils effectively [41].

Finally, when schools prioritize mental health, they contribute to healthier communities overall [42]. Schools are the primary setting for social interaction and relationship-building among peers. They play a critical role in fostering healthy social skills, empathy, and emotional intelligence, which are essential for mental well-being. Mentally healthy pupils and educators are more likely to engage in positive relationships, participate in community activities, and contribute to society’s well-being [43].

Policy implications summarized in the mental health literature include the selection of indicators of student mental health and well-being; incentivizing teaching education programs to include mental health literacy; establishing mental health as a component of curricula; providing funding for local education and behavioral health authorities to increase mental health awareness and promotion in schools, including early identification, intervention, and treatment in schools; and maximizing reimbursement for school mental health services [26].

### 3.1. Integration of Mental Health Education in Core Curricula

Through the integration of mental health education into core curricula, schools may increase mental health literacy among both educators and pupils [44]. Better health literacy may equip children and adolescents with vital coping skills, foster emotional intelligence, and support their overall well-being, ultimately promoting a healthier and more successful future.

There are several reasons for integrating mental health education in school curricula in times of crisis [44,45,46,47]:The reduction in stigma helps normalize conversations around mental health, reduce stigma, and encourage open dialogue among students, teachers, and parents.Mental health education can help students understand their emotions, thoughts, and behaviors. Self-awareness is essential for recognizing and managing mental health needs.Curricula might support building coping skills through strategies for managing stress, anxiety, and other emotions. This skill is crucial for building healthy relationships and for navigating social situations.Providing mental health education early on can help prevent or mitigate mental health challenges by teaching students how to recognize warning signs and seek help when needed.

Several steps are recommended to effectively integrate mental health education into the curriculum and achieve these goals (Table 1).

Differentiating interventions based on the age of beneficiaries is crucial for the effective implementation of mental health support. For instance, younger children may benefit more from programs that focus on building emotional literacy, self-awareness, and basic coping strategies, whereas interventions for adolescents could delve deeper into topics such as stress management, understanding complex emotions, and navigating social and academic pressures. The content of the interventions should evolve as pupils grow and their cognitive and emotional capacities expand [48].

The type of stressors that pupils face can also influence the design of interventions. For example, children and adolescents experiencing acute stressors, such as a family crisis or traumatic event, might require more intensive and individualized support, possibly involving therapeutic interventions. Conversely, those dealing with chronic stressors such as academic pressure or social-media-related anxiety could benefit from group-based programs that teach coping strategies and resilience skills.

Finally, geographical context plays a role in shaping interventions. Schools in urban areas might need to address issues such as gang violence or drug abuse, whereas those in rural areas could focus on challenges such as social isolation or limited access to mental health services. Schools in regions affected by conflicts or natural disasters would need to provide trauma-informed care and possibly collaborate with local community organizations and policymakers to provide comprehensive support.

Training educators in recognizing early signs of mental health issues and strategies for intervention can be achieved through professional development workshops led by mental health professionals. An example of a specific program could be Mental Health First Aid, a program designed to equip educators with the skills to respond to signs of mental health issues in students [56].

Supportive environment can be created by an atmosphere that encourages open dialogue about mental health by implementing programs such as “Circle Time” or “Advisory Periods,” where students are given a safe space to express their feelings and discuss mental health topics [57,58,59]. These discussions could be guided by a mental health professional or a trained staff member to ensure that they are conducted in a supportive and respectful manner. As for offering resources, schools could provide mental health resources through a dedicated section in the school library or an online portal where students can find information about different mental health conditions, coping strategies, and local mental health services. An example of such a resource could be the “Mindfulness in Schools Project,” which provides mindfulness resources and training for schools [60]. Schools could establish a referral system for mental health services to support students on their mental health journey. This could involve identifying students in need, providing them with information about available mental health services, facilitating access to these services.

Schools may face several challenges in implementing these tasks, such as limited resources, a lack of trained staff, and varying levels of acceptance and understanding of mental health issues among students, parents, and educators. Strategies to navigate these challenges could include seeking partnerships with local mental health organizations, applying for grants or other funding opportunities to resource mental health initiatives, and conducting regular training and awareness-raising sessions for all stakeholders.

Schools can employ a multifaceted evaluation approach to measure the progress of these tasks. For example, surveys could be administered to students, staff, and parents to gauge changes in attitudes towards mental health, awareness of mental health resources, and the perceived effectiveness of mental health support. Furthermore, schools could track referral rates to mental health services, participation rates in mental health programs, and changes in student well-being indicators such as attendance rates, academic performance, and behavioral issues.

#### Context-Specific Interventions for Mental Health Support

Specific context-based interventions related to the unique challenges of the COVID-19 pandemic, climate change, and geopolitical conflicts must be developed and applied.

The global health crisis caused by the COVID-19 pandemic has caused significant disruptions to education and normal life routines, leading to increased stress, anxiety, and feelings of isolation among pupils. Age-appropriate interventions include virtual support groups, online mental health resources, and resilience-building activities [61]. For younger children, creative outlets such as art and play therapy can be useful in expressing emotions and managing stress [62]. For adolescents, online peer support platforms and cognitive behavioral therapy techniques can help them cope with the unique stressors associated with the pandemic [63].The growing prevalence of “eco-anxiety” can be addressed by integrating climate education into the curriculum, encouraging eco-friendly practices, and facilitating open discussions about climate-related fears and emotions. Younger children can engage in nature-based play and learning to foster a connection with the environment [64], while adolescents can participate in climate action projects, promoting empowerment and resilience [65].Children and adolescents in conflict-affected areas face unique mental health challenges owing to trauma and instability. Here, trauma-informed care, peace education, and conflict-resolution programs are crucial [66]. Young children may benefit from psychosocial support through play and art therapy [67], while older adolescents may need access to counseling and programs that build resilience and coping skills [68].

### 3.2. Interdisciplinary Collaboration in Mental Health Support of Children’s and Adolescents

Prioritizing mental health in schools requires interdisciplinary collaboration among mental health professionals, community organizations, and policymakers [26,69]. The relevant tasks in the mental health support of pupils are presented in Table 2. In summary, interdisciplinary collaboration among mental health professionals, community organizations, and policymakers is crucial for crafting and implementing effective support strategies in schools. This collaborative approach allows for a comprehensive understanding of mental health issues, leverages the unique expertise and resources of different stakeholders, and promotes tailored interventions, continuity of care, policy changes, stigma reduction, and ongoing evaluation and improvement.

## 4. Discussion

To develop and implement targeted interventions for children and adolescents, working with schools, mental health professionals, community organizations, and policymakers is necessary [77]. Collaboration ensures that the interventions are well-rounded and reach the intended audience effectively. Regular evaluations of the effectiveness of these interventions and adjustments based on feedback, and observed outcomes are recommended to ensure that resources are used efficiently and that interventions have the desired impact.

The increase in mental health problems among students is a complex issue, and it cannot be attributed solely to the failure of prevention efforts or an increase in stressors [78]. Several factors may contribute to the increase in mental health issues among students. Over the years, there has been growing awareness and understanding of mental health issues. This increased awareness may lead to higher reporting rates, giving the impression of a rise in mental health problems when they are simply identified and documented more effectively [79].

Children and adolescents face unique stressors that may not have been present or prevalent in previous generations. These include increased academic pressure, the impact of social media, and an uncertain job market. These evolving stressors may contribute to the rise in mental health issues among the youth. Broader societal changes, such as economic instability, political polarization, and global issues, such as climate change, can also impact the mental health of young people, adding to the stress and anxiety they experience. The widespread use of technology and social media have both positive and negative effects on mental health [80]. Although it can provide a means of connection and support, it can also contribute to feelings of isolation, low self-esteem, and cyberbullying. Socioeconomic disparities and systemic inequalities can play a significant role in mental health outcomes. Students from disadvantaged backgrounds may experience higher levels of stress and adversity and have limited access to mental health resources.

Although prevention efforts have been implemented in many schools, there may still be a lack of adequate resources, support, and training for educators and mental health professionals. This could hinder the effectiveness of prevention strategies and contribute to an increase in mental health issues [81].

Policy modifications within educational institutions play a pivotal role in mitigating students’ multi-faceted mental health issues. Such modifications should include an explicit focus on mental health in the curriculum. Concrete modules dedicated to understanding mental health conditions and the importance of self-care, resilience, and healthy coping strategies can be introduced. Furthermore, integrating pertinent topics such as climate change awareness, conflict resolution, and financial literacy will help equip students with an understanding of global stressors and methods to manage their impact on mental health.

While the institutionalization of mental health professionals within schools is becoming increasingly common, a stronger focus on defining their roles and creating clear referral pathways for external mental health services can further enhance the support system. Schools should also consider nurturing partnerships with local community organizations to leverage their resources and expertise. Inviting local environmental, financial, and mental health organizations to conduct workshops and programs can create an engaging, supportive, and informed school environment.

Assessments and disciplinary procedures should reflect a school’s commitment to mental health support. Acknowledging signs of mental distress in academic performance and behavior, and adopting supportive rather than punitive measures in response, can help cultivate an environment that prioritizes student well-being. This aligns with a broader societal shift towards understanding and addressing mental health concerns and the recognition that mental health is a crucial component of overall well-being.

By embedding these strategies within their policies, educational institutions can ensure that mental health support is not an add-on but rather an integral part of their commitment to nurturing resilient, well-equipped students.

## 5. Conclusions

The increase in mental health problems among students is likely due to a combination of factors including increased awareness, evolving stressors, societal changes, inadequate resources, persistent stigma, the role of technology, and inequality. Mental health must become a priority in schools, as reflected in the adaptation of the curriculum according to modern didactic guidelines. The prevention, early detection, and treatment of mental health problems should occur more intensively and comprehensively in schools. This study stresses the significance of continuously evaluating and refining mental health support policies to ensure their efficacy and adaptability in the face of evolving crises. Addressing these complex and interconnected factors will require ongoing efforts and collaboration among educators, mental health professionals, policymakers, and communities. Educational institutions can effectively leverage the expertise of diverse stakeholders to create targeted interventions by cultivating partnerships. 

In light of the complexities surrounding the increasing prevalence of mental health issues among students, we recommend adopting a multi-faceted approach in schools. This approach should encompass the integration of mental health education into the core curriculum, comprehensive training of educators and school staff in mental health awareness and support, and the establishment of dedicated roles for mental health professionals within the school setting. Additionally, schools should actively seek partnerships with local community organizations and policymakers to ensure a broad-based and effective mental health support system.

In terms of policy, we recommend that educational institutions implement policies that prioritize mental wellbeing in their assessment and disciplinary procedures, recognize signs of mental distress as a factor in academic performance and behavior, and respond with supportive measures.

Finally, we recommend that schools adopt a continuous evaluation and refinement process for their mental health support strategies to ensure their efficacy and adaptability in the face of evolving crises and changing student needs. By implementing these recommendations, schools can help foster a more supportive, understandable, and resilient environment for students.

## 6. Future Directions

Through comprehensive recommendations on the integration of mental health education into core curricula and interdisciplinary collaboration in the mental health support of children and adolescents in schools, the opinion article sought to catalyze a transformation in educational policies, prioritize mental health support, and empower pupils to thrive during tumultuous times.

While our proposal outlined a comprehensive and multifaceted approach to mental health support within educational systems, several potential limitations and obstacles must be acknowledged. First, implementing mental health education into core curricula and training educators requires significant financial and time resources. Schools, especially those in under-resourced areas, may face challenges in securing these resources. Second, the stigma surrounding mental health may present a barrier to open dialogue and the acceptance of these initiatives among students, staff, and the broader community. Even with the best educational interventions, stigma can prevent those who need help from seeking help. Third, the diversity of student experiences and needs calls for tailored interventions. A one-size-fits-all approach to mental health education may not adequately address the specific issues faced by different student populations, such as those from diverse socioeconomic backgrounds, those with special needs, or those affected by unique stressors, such as conflict or displacement. Fourth, while interdisciplinary collaboration is crucial, it can also be challenging to coordinate efforts among different stakeholders, including educators, mental health professionals, policymakers, and community organizations. Finally, evaluating the effectiveness of mental health interventions is complex, as mental health outcomes can be influenced by a variety of factors, some of which may be outside the school’s control. Despite these potential obstacles, the benefits of integrating mental health support into educational systems far outweigh these challenges, and there is reason to be optimistic about the transformative potential of these initiatives.

## Figures and Tables

**Table 1 healthcare-11-01423-t001:** Integration of mental health education into core curricula.

Task	Description
Develop age-appropriate curricula [48]	Mental health education should be tailored to the developmental needs of pupil at different ages, ensuring the content is relevant and engaging
Train educators [49,50,51]	Teachers should be trained to deliver mental health education, recognize warning signs, and provide support or referrals when needed
Foster a supportive environment [36]	Schools should create an atmosphere that encourages open dialogue about mental health, offers resources, and supports students in their mental health journey
Encourage parental involvement [52,53,54]	Parents play a crucial role in supporting their child’s mental health. Schools can involve parents by offering resources, workshops, and regular communication to keep them informed and engaged
Use interactive and engaging teaching methods [54,55]	Mental health education should incorporate various teaching methods, such as role-playing, group discussions, and multimedia resources, to keep students engaged and facilitate learning
Focus on social–emotional learning [26]	Integrating social–emotional learning principles into the curriculum can help pupils develop self-awareness, self-management, social awareness, relationship skills, and responsible decision-making
Evaluate and update curricula [26]	Regularly assess the effectiveness of mental health education and update the curriculum to ensure it remains relevant and responsive to the needs of students

**Table 2 healthcare-11-01423-t002:** Interdisciplinary collaboration for mental health support of pupils.

Item	Description
Comprehensive understanding [70,71]	Mental health is a multifaceted issue that intersects with various aspects of a pupil’s life, including their academic, social, and emotional development. Interdisciplinary collaboration allows for a comprehensive understanding of these factors and helps create well-rounded support strategies
Expertise and resources [72]	Each stakeholder brings unique expertise and resources to the table. General practitioners and mental health professionals offer clinical knowledge and intervention strategies, community organizations provide local context and additional support, and policymakers can create supportive legislation and allocate funding
Tailored interventions [73]	Collaboration between different stakeholders enables the development of tailored interventions that consider the specific needs, resources, and cultural factors of a school or community. This ensures that support strategies are more effective and better suited to the target population
Continuity of care [74]	Schools are just one part of a student’s support network. By collaborating with external organizations and professionals, schools can ensure a continuity of care for students both within and outside of the educational setting. This holistic approach helps address the root causes of mental health issues and supports long-term well-being
Policy and advocacy [26]	Policymakers play a critical role in creating a supportive environment for mental health initiatives in schools. By working together, stakeholders can advocate for policy changes that promote mental health education, allocate resources, and ensure that schools prioritize mental well-being
Reducing stigma [75]	Collaborative efforts among various stakeholders can help raise awareness about mental health issues and reduce stigma. This creates a more supportive culture where students feel comfortable seeking help and discussing their mental health concerns
Evaluation and improvement [76]	Interdisciplinary collaboration enables stakeholders to monitor and evaluate the effectiveness of support strategies, identify areas for improvement, and share best practices. This continuous feedback loop ensures that mental health initiatives in schools are constantly evolving and improving to better serve students’ needs

## Data Availability

No new data were created.

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
