# Peer review of "Fortifying the Foundations: A Comprehensive Approach to Enhancing Mental Health Support in Educational Policies Amidst Crises"

_healthcare, 2023, doi:10.3390/healthcare11101423_

Round 1

Reviewer 1 Report

Good originality, good contribution to the field, good technical quality, good clarity of presentation, good depth of research.

General Comment: an interesting work. It is well written and does tell a interesting interpretation to the reader. The topic is interesting in general.

Author(s) have studied and used an appropriate number of bibliography sources. The aim and background of the research problem are clearly described. Methodology of the research is clear. The applied methods and the interpretation and presentation of results correspond to international standards. But, it's missing a key methodological point: I cannot see research problem and the description of the research procedure. It is obligatory to add it to the methodology.

Author(s) provides original results of their investigations and examination of material from their own collections.

Reviewer 2 Report

This opinion piece discusses the need for education systems to prioritize students' mental health through specific tasks, outlined in the article. This well-articulated article does a nice job explaining the importance of this topic through the prevalence of mental health issues that students face and cites relevant literature to support these tasks. However, I expected the article would have talked through specific strategies that schools could adapt to carry out the tasks. The list of tasks seems idealistic without grounded steps that would guide schools towards realistic goals. For example, "train educators" is vague concept that could take many directions. I recommend adding resources or providing examples that would ground these tasks in more tangible concepts. I also recommend acknowledging the caveats schools face when approaching these tasks, and providing a way to measure the progress of these tasks to denote change in the school system. The discussion vaguely reemphasizes the points made in the introduction, but could provide more depth towards directions schools can take to create or change educational policies. The abstract refers to "comprehensive recommendations," but I believe revision is needed for these recommendations to land on a stronger foundation. 

Reviewer 3 Report

Good survey paper of exiting articles advocating the topic. A clear recommendation is missing. Additional literature references that may e relevant: 1) “Schools and Disasters: Safety and Mental Health Assessment and Interventions for Children, ” published in Current Psychiatry Reports https://doi.org/10.1007/s11920-016-0743-9  and 2) Educational and psychosocial support for conflict-affected youths: The effectiveness of a school-based intervention targeting academic underachievement https://doi.org/10.1080/21683603.2022.2043209

Reviewer 4 Report

The contribute suggests a pedagogical methodology whose effectiveness has been strongly supported over time in variety of fields, such as the Flipped Classroom.

However, each stage of the life cycle needs an appropriate approach. Therefore, I would invite you to emphasize in paragraph 2.1 (e.g., by briefly specifying in the first step of Table 1. Integration of mental health education into core curricula, or within your discussion), the differences in interventions in relation to the age of the beneficiaries (children or adolescents), or to the stressors typology, or to geographical context. 

In the present drafting, it seems to emerge a persistent danger in presenting all these suggestions (provided within the text) in such very general view.

On the contrary, I suggest specifically to combine interventions focused to age-target beneficiaries, writing on the specific problem context which have significantly impacted the mental health (COVID-19 pandemic, or climate change, or geopolitical conflicts- blindly repeated all over the article).

Moreover, the number of key words appear to be excessive (it is better to enlist four/five key words).

Finally, within paragraph 5.5 I would suggest to write possible limitations or obstacles to your proposal.

In conclusion, along the text do not appear any linking words (lines 36; 38; 39; 41; 43 etc). For istance, it should be suggested a revision of English language (through a native speaker).

Round 2

Reviewer 4 Report

I appreciate the author's efforts, and the manuscript has enjoyed the insights